

# NFAD: fixing anomaly detection using normalizing flows

Artem Ryzhikov, Maxim Borisyak, Andrey Ustyuzhanin and Denis Derkach

Laboratory of Methods for Big Data Analysis, HSE University, Moscow, Russia

## ABSTRACT

Anomaly detection is a challenging task that frequently arises in practically all areas of industry and science, from fraud detection and data quality monitoring to finding rare cases of diseases and searching for new physics. Most of the conventional approaches to anomaly detection, such as one-class SVM and Robust Auto-Encoder, are one-class classification methods, *i.e.*, focus on separating normal data from the rest of the space. Such methods are based on the assumption of separability of normal and anomalous classes, and subsequently do not take into account any available samples of anomalies. Nonetheless, in practical settings, some anomalous samples are often available; however, usually in amounts far lower than required for a balanced classification task, and the separability assumption might not always hold. This leads to an important task—incorporating known anomalous samples into training procedures of anomaly detection models. In this work, we propose a novel model-agnostic training procedure to address this task. We reformulate one-class classification as a binary classification problem with normal data being distinguished from pseudo-anomalous samples. The pseudo-anomalous samples are drawn from low-density regions of a normalizing flow model by feeding tails of the latent distribution into the model. Such an approach allows to easily include known anomalies into the training process of an arbitrary classifier. We demonstrate that our approach shows comparable performance on one-class problems, and, most importantly, achieves comparable or superior results on tasks with variable amounts of known anomalies.

Corresponding author
Artem Ryzhikov, aryzhikov@hse.ru

## INTRODUCTION

The anomaly detection (AD) problem is one of the important tasks in the analysis of real-world data. Possible applications range from the data-quality certification (for example, *Borisyak et al., 2017*) to finding the rare specific cases of the diseases in medicine (*Spence, Parra & Sajda, 2001*). The technique can be also used in credit card fraud detection (*Aleskerov, Freisleben & Rao, 1997*), complex systems failure predictions (*Xu & Li, 2013*), and novelty detection in time series data (*Schmidt & Simic, 2019*).

Formally, AD is a classification problem with a representative set of normal samples and a small, non-representative or empty set of anomalous examples. Such a setting makes conventional binary classification methods to be overfitted and not to be robust w.r.t. novel anomalies (*Görnitz et al., 2012*). In contrast, conventional one-class classification (OC-)

methods (*Breunig et al., 2000*; *Liu, Ting & Zhou, 2012*) are typically robust against all types of outliers. However, OC-methods do not take into account known anomalies which often result to suboptimal performance in cases when normal and anomalous classes are not perfectly separable (*Campos et al., 2016*; *Pang, Shen & Van den Hengel, 2019*). The research in the area addresses several challenges (*Pang et al., 2021*) that lie in the field of increasing precision, generalizing to unknown anomaly classes, and tackling multi-dimensional data. Several reviews of classical (*Zimek, Schubert & Kriegel, 2012*; *Aggarwal, 2016*; *Boukerche, Zheng & Alfandi, 2020*; *Belhadi et al., 2020*) and deep-learning methods (*Pang et al., 2021*) were published that describe the literature in detail. With the advancement of the neural generative modeling, methods based on generative adversarial networks (*Schlegl et al., 2017*), variational autoencoders (*Xu et al., 2018*), and normalizing flows (*Pathak, 2019*) are introduced for the AD task.

We propose[1] addressing the class-imbalanced classification task by modifying the learning procedure that effectively makes anomaly detection methods suitable for a two-class classification. Our approach relies on imbalanced dataset augmentation by surrogate anomalies sampled from normalizing flow-based generative models.

## PROBLEM STATEMENT

Classical AD methods consider anomalies a priori significantly different from the normal samples (*Aggarwal, 2016*). In practice, while such samples are, indeed, most likely to be anomalous, often some anomalies might not be distinguishable from normal samples (*Hunziker et al., 2017*; *Pol et al., 2019*; *Borisyak et al., 2017*). This provides a strong motivation to include known anomalous samples into the training procedure to improve the performance of the model on these ambiguous samples. Technically, this leads to a binary classification problem which is typically solved by minimizing cross-entropy loss function $L_{BCE}$:

$$f^*(x) = \arg \min_f L_{BCE}(f);$$ (1)

$$L_{BCE}(f) = P(C^+)\mathbb{E}_{x \sim C^+}\log f(x) + P(C^-)\mathbb{E}_{x \sim C^-}\log\big(1 - f(x)\big);$$ (2)

where: $f$ is a arbitrary model (*e.g.*, a neural network), $C^+$ and $C^-$ denote normal and anomalous classes. In this case, the solution $f^*$ approaches the optimal Bayesian classifier:

$$f^*(x) = P(C^+|x) = \frac{p(x|C^+)p(C^+)}{p(x|C^+)p(C^+) + p(x|C^-)p(C^-)}.$$ (3)

Notice that $f^*$ implicitly relies on the estimation of the probability densities $P(x|C^+)$ and $P(x|C^-)$. A good estimation of these densities is possible only when a sufficiently large and representative sample is available for each class. In practical settings, this assumption certainly holds for the normal class. However, the anomalous dataset is rarely large or representative, often consisting of only a few samples or covering only a portion of all possible anomaly types.[2] With only a small number of examples (or a non-representative sample) to estimate the second term of Eq. (2), $L_{BCE}$ effectively does not depend on $f(x)$

[1] Source code is available at https://gitlab.com/lambda-hse/nfad.

[2] Moreover, anomalies typically cover a much larger "phase space" than normal samples, thus, generic models (*e.g.*, a deep neural network with fully connected layers) might require significantly more anomalous examples than normal ones.

in $x \in \text{supp}\,C^- \setminus \text{supp}\,C^+$, which leads to solutions with arbitrary predictions in the area, *i.e.,* to classifiers that are not robust to novel anomalies.

One-class classifiers avoid this problem by aiming to explicitly separate the normal class from the rest of the space (*Liu, Ting & Zhou, 2008*; *Scholkopf & Smola, 2018*). As discussed above, this approach, however, ignores available anomalous samples, potentially leading to incorrect predictions on ambiguous samples.

Recently, semi-supervised AD algorithms like $1+\varepsilon$-classification method (*Borisyak et al., 2020*), Deep Semi-supervised AD method (*Ruff et al., 2019*), Feature Encoding with AutoEncoders for Weakly-supervised Anomaly Detection (*Zhou et al., 2021*) and Deep Weakly-supervised Anomaly Detection (*Pang et al., 2019*) were put forward. They aim to combine the main properties of both unsupervised (one-class) and supervised (binary classification) approaches: proper posterior probability estimations of binary classification and robustness against novel anomalies of one-class classification.

In this work, we propose a method that extends the $1+\varepsilon$-classification method (*Borisyak et al., 2020*) scheme by exploiting normalizing flows. The method is based on sampling the surrogate anomalies to augment the existing anomalies dataset using advanced techniques.

## NORMALIZING FLOWS

The normalizing flows (*Rezende & Mohamed, 2015b*) generative model aims to fit the exact probability distribution of data. It represents a set of invertible transformations $\{f_i(\cdot\,;\theta_i)\}$ with parameters $\theta_i$, to obtain a bijection between the given distribution of training samples and some domain distribution with known probability density function(PDF). However, in the case of non-trivial bijection $z_0 \leftrightarrow z_k$, the distribution density at the final point $z_k$ (training sample) differs from the density at point $z_0$ (domain). This is due to the fact that each non-trivial transformation $f_i(\cdot\,;\theta_i)$ changes the infinitesimal volume at some points. Thus, the task is not only to find a flow of invertible transformations $\{f_i(\cdot\,;\theta_i)\}$, but also to know how the distribution density is changed at each point after each transformation $f_i(\cdot\,;\theta_i)$.

Consider the multivariate transformation of variable $z_i = f_i(z_{i-1};\theta_i)$ with parameters $\theta_i$ for $i > 0$. Then, Jacobian for a given transformation $f_i(z_{i-1};\theta_i)$ at given point $z_{i-1}$ has the following form:

$$J(f_i|z_{i-1}) = \begin{bmatrix} \dfrac{\partial f_i}{\partial z_{i-1}^1} & \cdots & \dfrac{\partial f_i}{\partial z_{i-1}^n} \end{bmatrix} = \begin{bmatrix} \dfrac{\partial f_{i-1}^1}{\partial z_{i-1}^1} & \cdots & \dfrac{\partial f_{i-1}^1}{\partial z_{i-1}^n} \\ \vdots & \ddots & \vdots \\ \dfrac{\partial f_{i-1}^m}{\partial z_{i-1}^1} & \cdots & \dfrac{\partial f_{i-1}^m}{\partial z_{i-1}^n} \end{bmatrix} \tag{4}$$

Then, the distribution density at point $z_i$ after the transformation $f_i$ of point $z_{i-1}$ can be written in a following common way:

$$p(z_i) = \frac{p(z_{i-1})}{|\det J(f_i|z_{i-1})|}, \tag{5}$$

where $\det J(f_i|z_{i-1})$ is a determinant of the Jacobian matrix $J(f_i|z_{i-1})$ (*Rezende & Mohamed, 2015*).

Thus, given a flow of invertible transformations $\boldsymbol{f} = \{f_i(\cdot;\theta_i)\}_{i=1}^N$ with known $\{\det J(\boldsymbol{f}_i|\cdot)\}_{i=1}^N$ and domain distribution of $\boldsymbol{z}_0$ with known p.d.f. $p(\boldsymbol{z}_0)$, we obtain likelihood $p(\boldsymbol{x})$ for each object $\boldsymbol{x} = \boldsymbol{z}_N$. This way, the parameters $\{\theta_i\}_{i=1}^N$ of NF model $\boldsymbol{f}$ can be fitted by explicit maximizing the likelihood $p(\boldsymbol{x})$ for training objects $\boldsymbol{x} \in X$. In practice, Monte-Carlo estimate of $\log p(X) = \log \Pi_{x \in X} p(x) = \Sigma_{x \in X} \log p(x)$ is optimized, which is an equivalent optimization procedure. Also, the likelihood $p(X)$ can be used as a metric of how well the NF model $\boldsymbol{f}$ fits given data $X$.

The main bottleneck of that scheme is located in that $\det J(\cdot|\cdot)$ computation, which is $O(n^3)$ in a common case ($n$ is the dimension of variable $\boldsymbol{z}$). In order to deal with that problem, specific normalizing flows with specific families of transformations $\boldsymbol{f}$ are used, for which Jacobian computation is much faster (*Rezende & Mohamed, 2015*; *Papamakarios, Pavlakou & Murray, 2017*; *Kingma et al., 2016*; *Chen et al., 2019*).

## ALGORITHM

The suggested NF-based AD method (NFAD) is a two-step procedure. In the first step, we train normalizing flow on normal samples to sample new surrogate anomalies. Here, we assume that anomalies differ from normal samples, and its likelihood $p_{NF}(x^-|C^+)$ is less than likelihood of normal samples $p_{NF}(x^+|C^+)$. In the second step, we sample new surrogate anomalies from tails of normal samples distribution using NF and train an arbitrary binary classifier on normal samples and a mixture of real and sampled surrogate anomalies.

### Step 1. Training normalizing flow

We train normalizing flow on normal samples. It can be trained by a standard for normalizing flows scheme of maximization the log-likelihood (see 'Normalizing flows'):

$$\max_{\theta} L_{NF} \tag{6}$$

$$L_{NF} = \mathbb{E}_{x \sim C^+} \log p_f(x) \tag{7}$$

$$= \mathbb{E}_{z \sim f^{-1}(C^+;\theta)} \left[ \log p(z) - \log |\det J(f|z)| \right], \tag{8}$$

where $f(\cdot;\theta)$ is NF transformation with parameters $\theta$, $J(f|z)$ is Jacobian of transformation $f(z;\theta)$ at point $z$, $z$ are samples from multivariate standard normal domain distribution $p(z) = \mathcal{N}(z|0,I)$, $x$ are normal samples from the training dataset, $p_f(x) = \frac{p(z)}{J(f|z)}\big|_{z=f^{-1}(x;\theta)}$.

After NF for sampling is trained, it can be used to sample new anomalies. To produce new anomalies, we sample $z$ from tails of normal domain distribution, where $p$-value of tails is a hyperparameter (see Fig. 1).

Here, we assume that test time anomalies are either represented in the given anomalous training set or novelties w.r.t. normal class. In other words, $p(x|C^+)$ of novelties $x$ must be relatively small. Nevertheless, $p(x)$ obtained by NF might be drastically different from

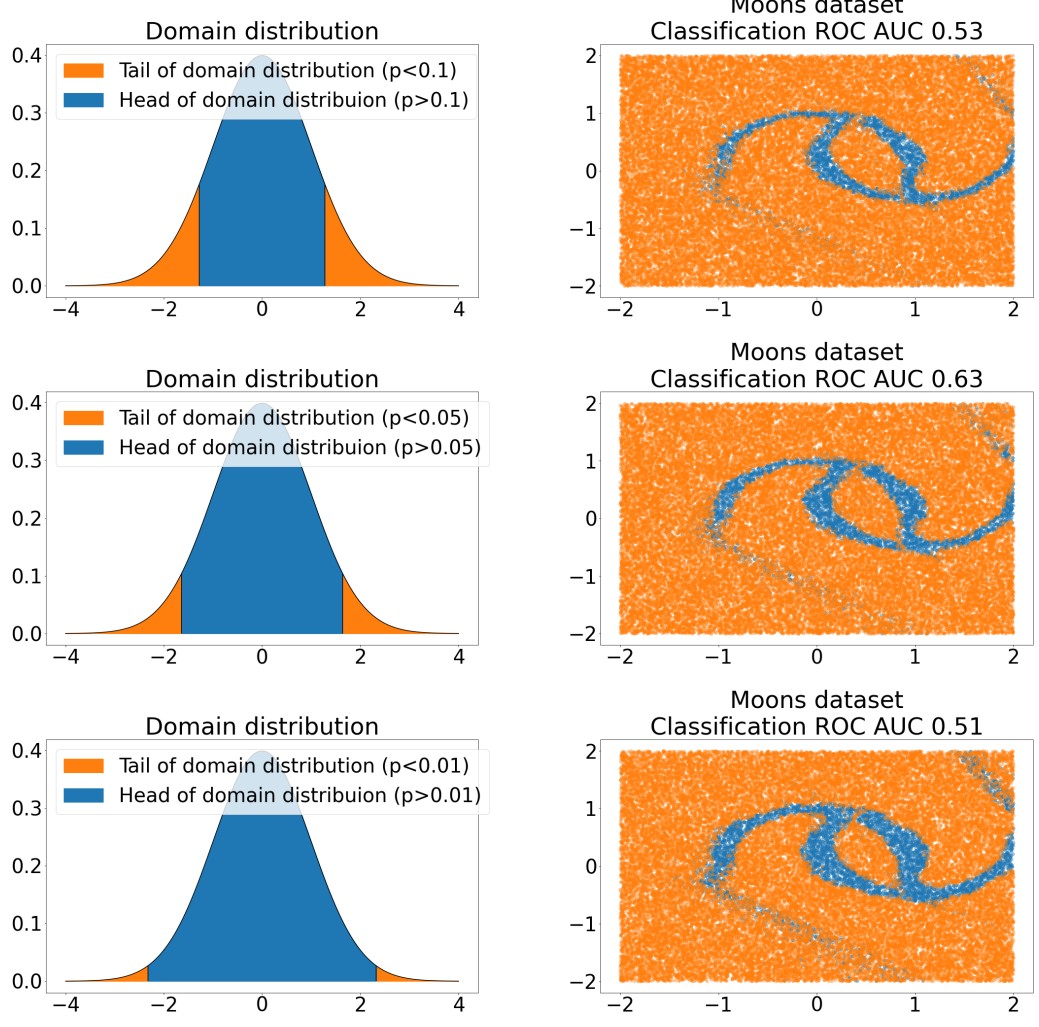

**Figure 1** **NF bijection between tails of standard normal domain distribution (left) and 2D Moon dataset (*Pedregosa et al., 2011*) samples (right).** Rows represent different tail *p*-values choices. The value of the ROC AUC of the anomaly classifier is shown on the right side. The classifier is trained on the mixture of $C^+$ samples from the Moon dataset and surrogate anomalies sampled from the tails.

the corresponding domain point likelihood $p(z)$ because of non-unit Jacobian of NF transformations Eq. (8). Such distribution density distortion is illustrated in Fig. 2 and makes the proposed sampling scheme of anomalies to be incomplete. Because of such distortion, some points in the tails of the domain can correspond to normal samples, and some points in the body of domain distribution can correspond to anomalies. To fix it, we propose Jacobian regularization of normalizing flows (Fig. 2) by introducing extra regularization term. It penalizes the model for non-unit Jacobian:

$$L_J = \mathbb{E}_{z \sim \mathcal{N}(0,1)} \log(|\det J(f|z)|)^2 \tag{9}$$

$$\max_\theta \left[ L_{NF} - \lambda * L_J \right], \lambda \geq 0, \tag{10}$$

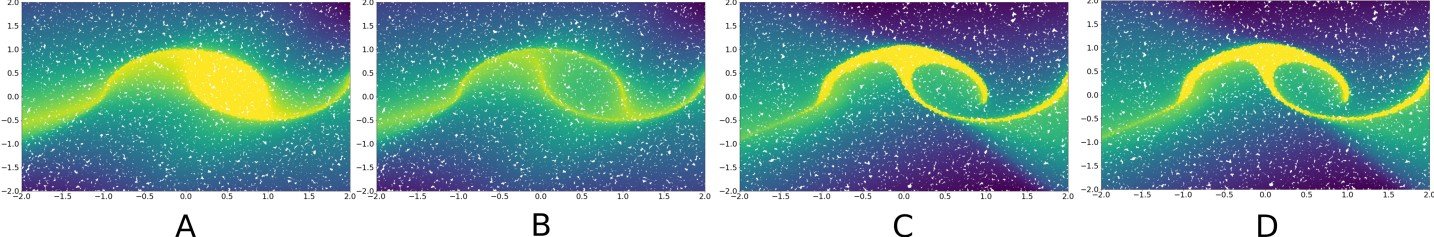

**Figure 2 Density distortion of normalizing flows on the Moon dataset (*Pedregosa et al., 2011*).** Without extra regularization distribution density of domain distribution (A) significantly differs from the target distribution (B) because of non-unit Jacobian. To preserve the distribution density after NF transformations, Jacobian regularization Eq. (9) can be used (C and D, respectively).

where $\lambda$ denotes the regularization hyperparameter. We estimate the regularization term $L_J$ in Eq. (9) by direct sampling of $z$ from the domain distribution $\mathcal{N}(0, I)$ to cover the whole sampling space. The theorem below proofs that any level of expected distortion can be obtained with such a regularization:

**Theorem 4.1** *Let $\Omega \subset \mathbb{R}^d$ a sample space with probability (domain) distribution $\mathcal{D}$, $C^+ \subset \Omega$ a class of normal samples, $f(\cdot; \theta) : \mathbb{R}^d \to \mathbb{R}^d$ is a set of invertible transformations parametrized by $\theta$ and $\exists \theta_0 : f(\cdot; \theta_0) = $ (identical transformation exists). Then $I \forall \varepsilon > 0$ $\exists \lambda \geq 0$ such that $[E_{z \sim \mathcal{D}} \log(|\det J(f|z)|)^2]_{\theta*} < \varepsilon \mid \forall z \sim \Omega \in \mathbb{R}^d$, where $\theta* \in \arg\min_\theta \big[ - \mathbb{E}_{x \sim C^+} \log p_f(x) + \lambda \mathbb{E}_{z \sim \mathcal{D}} \log(|\det J(f|z)|)^2 \big], p_f(x) = \frac{p(z)}{J(f|z)}\big|_{z = f^{-1}(x; \theta)}.$*

*Proof.* Suppose the opposite. Let $\exists \varepsilon > 0$ s.t. $\forall \lambda \geq 0 : \big[ \mathbb{E}_{z \sim \mathcal{D}} \log(|J(f|z)|)^2 \big]_{\theta*} \geq \varepsilon$ for all $\theta* \in \arg\min_\theta \big[ - \mathbb{E}_{x \sim C^+} \log p_f(x) + \lambda \mathbb{E}_{z \sim \mathcal{D}} \log(|\det J(f|z)|)^2 \big]$.

Since $\exists \theta_0 : f(\cdot; \theta_0) = I, p_f(f(z; \theta_0)) = p(z) \big| \forall z \sim \Omega$, the term

$$\big[ \mathbb{E}_{z \sim \mathcal{D}} \log(|\det J(f|z)|)^2 \big]_{\theta_0} = 0$$

since

$$\frac{p(z)}{p(f(z; \theta_0))} = |\det J(f|z)|_{\theta_0} = 1 \big| \forall z \in \Omega$$

Let $\big[ - \mathbb{E}_{x \sim C^+} \log p_f(x) \big]_{\theta_0} = - \mathbb{E}_{z \sim C^+} \log p(z) = c_0, \min_\theta \big[ - \mathbb{E}_{x \sim C^+} \log p_f(x) \big] = c_{min} < c_0$ (minimum exists since negative log likelihood is lower bounded by 0). Then $\forall \lambda$:

$$c_0 > c_{min} + \lambda \big[ \mathbb{E}_{z \sim \mathcal{D}} \log(|\det J(f|z)|)^2 \big]_{\theta*} \geq c_{min} + \lambda \varepsilon$$

But $\lambda > \frac{c_0 - c_{min}}{\varepsilon}$ leads to contradiction. $\square$

In this work, we use Neural Spline Flows (NSF, *Durkan et al., 2019*) and Inverse (IAF, *Kingma et al., 2016*) Autoregressive Flows for tabular anomalies sampling. We also use Residual Flow (ResFlow, *Chen et al., 2019*) for anomalies sampling on image datasets. All the flows satisfy the conditions of Theorem 4.1. The proposed algorithms are called 'nfad-nsf', 'nfad-iaf' and 'nfad-resflow' respectively.

## Step 2. Training classifier

Once normalizing flow for anomaly sampling is trained, a classifier can be trained on normal samples and a mixture of real and surrogate anomalies sampled from NF (Fig. 3).

During the research, we used binary cross-entropy objective Eq. (2) to train the classifier. We do not focus on classifier configuration since any classification model can be used at this step.

### Final algorithm

The final scheme of the algorithm is shown in Fig. 3 accompanied with pseudocode Algorithm 1. All training details are given in Appendix A.

> **Input** : Normal samples $C^+$, anomaly samples $C^-$ (may be empty), $p$-value of tail $\boldsymbol{p}$, number of epochs for NF $E_{NF}$, number of epochs for classifier $E_{CLF}$
>
> **Output**: Anomalies classifier $g_\phi$
>
> **for** *epoch from 1 to $E_{NF}$* **do**
> > sample minibatch of normal samples $X^+ \sim C^+$;
> > calculate NF bijection between points on gaussian $Z^+$ and normal samples $X^+$: $Z^+ = f^{-1}(X^+; \theta)$;
> > update parameters $\theta$ of NF $f$ with the following gradient ascend: $\nabla_\theta \log p(X^+) = \nabla_\theta \big[ \log p(Z^+) - \log|\det J(f|Z^+)| \big]$;
>
> **end**
>
> **for** *epoch from 1 to $E_{CLF}$* **do**
> > sample $\tilde{Z}$ from gaussian tail: $\tilde{Z} \sim \mathcal{N}(0,1)$ s.t. $p(\tilde{Z}) \leq \boldsymbol{p}$;
> > sample surrogate anomalies $\tilde{X}$ using NF: $\tilde{X} = f(\tilde{Z}; \theta)$;
> > sample minibatch of normal samples: $X^+ \sim C^+$;
> > sample minibatch of anomalies (if $C^-$ is not empty): $X^- \sim C^-$;
> > update parameters $\phi$ of classifier $g_\phi$ with the following gradient descent: $\nabla_\phi \big[ \log g_\phi(X^+) + \log(1 - g_\phi(X^-)) + \log(1 - g_\phi(\tilde{X})) \big]$;
>
> **end**

**Algorithm 1:** NFAD algorithm

## RESULTS

We evaluate the proposed method on the following tabular and image datasets: KDD-99 (*Stolfo et al., 1999*), SUSY (*Whiteson, 2014*), HIGGS (*Baldi, Sadowski & Whiteson, 2014*), MNIST (*LeCun et al., 1998a*), Omniglot (*Lake, Salakhutdinov & Tenenbaum, 2015*) and CIFAR (*Krizhevsky, Hinton et al., 2009*). In order to reflect typical AD cases behind the approach, we derive multiple tasks from each dataset by varying sizes of anomalous datasets.

As the proposed method targets problems that are intermediate between one-class and two-class problems, we compare the proposed approach with the following algorithms:

- one-class methods: Robust AutoEncoder (RAE-OC, (*Chalapathy, Krishna Menon & Chawla, 2017*)) and Deep SVDD (*Ruff et al., 2018*).
- conventional two-class classification;
- semi-supervised methods: dimensionality reduction by an Deep AutoEncoder followed by two-class classification (DAE), Feature Encoding with AutoEncoders for Weakly-supervised Anomaly Detection (FEAWAD, (*Zhou et al., 2021*)), DevNet (*Pang, Shen &*

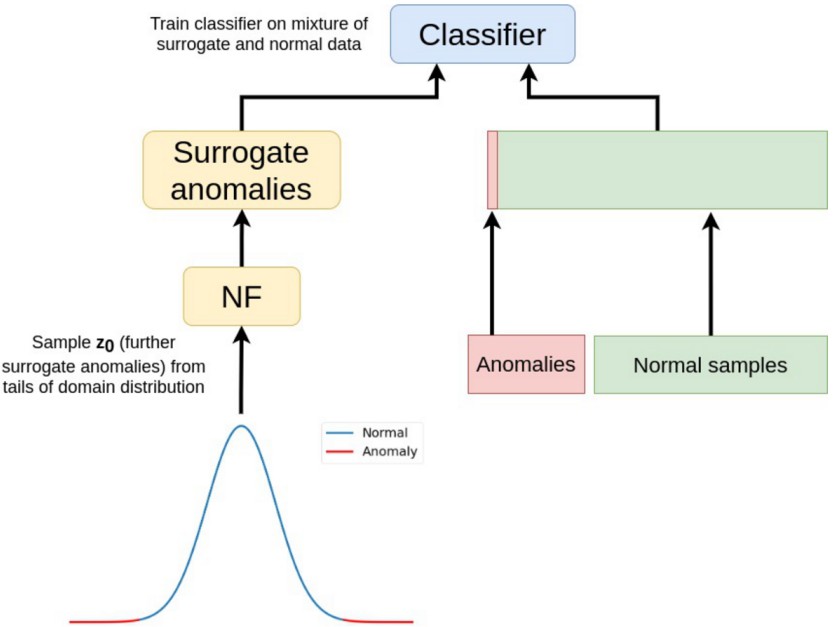

**Figure 3 Normalizing flows for anomaly detection (NFAD).** Surrogate anomalies are sampled from the tails of gaussian distribution and transformed by NF to be mixed into real samples. Then, any classifier can be trained on that mixture.

_Van den Hengel, 2019_), $1+\varepsilon$ method (_Borisyak et al., 2020_) ('*ope'), Deep SAD (_Ruff et al., 2019_) and Deep Weakly-supervised Anomaly Detection (PRO, (_Pang et al., 2019_))

We compare the algorithms using the ROC AUC metric to avoid unnecessary optimization for threshold-dependent metrics like accuracy, precision, or F1. Tables 1, 2 and 3 show the experimental results on tabular data. Tables 4, 5 and 6 show the experimental results on image data. Also, some of the aforementioned algorithms like DevNet are applicable only to tabular data and not reported on image data. In these tables, columns represent tasks with a varying number of negative samples presented in the training set: numbers in the header indicate either number of classes that form negative class (in case of KDD, CIFAR, OMNIGLOT and MNIST datasets) or a number of negative samples used (HIGGS and SUSY); 'one-class' denotes the absence of known anomalous samples. As one-class algorithms do not take into account negative samples, their results are identical for the tasks with any number of known anomalies. The best score in each column is highlighted in bold font.

## DISCUSSION

Our tests suggest that the best results are achieved when the normal class distribution has single mode and convex borders. These requirements are data-specific and can not be effectively addressed in our algorithm. The effects can be seen in Fig. 2, where two modes result in the "bridge" in the reconstructed standard class shape, and the non-convexity of the borders ends up in the worse separation line description.

**Table 1  ROC AUC on the KDD-99 dataset.** 'nfad*' is our algorithm.

|  | one class | 1 | 2 | 4 | 8 |
|---|---|---|---|---|---|
| rae-oc | 0.972 ± 0.006 | 0.972 ± 0.006 | 0.972 ± 0.006 | 0.972 ± 0.006 | 0.972 ± 0.006 |
| deep-svdd-oc | 0.939 ± 0.014 | 0.939 ± 0.014 | 0.939 ± 0.014 | 0.939 ± 0.014 | 0.939 ± 0.014 |
| two-class | – | 0.571 ± 0.213 | 0.700 ± 0.182 | 0.687 ± 0.268 | 0.619 ± 0.257 |
| dae | – | 0.685 ± 0.258 | 0.531 ± 0.286 | 0.758 ± 0.171 | 0.865 ± 0.087 |
| brute-force-ope | 0.564 ± 0.122 | 0.667 ± 0.175 | 0.606 ± 0.261 | 0.737 ± 0.187 | 0.541 ± 0.257 |
| hmc-eope | 0.739 ± 0.245 | 0.885 ± 0.152 | 0.919 ± 0.055 | 0.863 ± 0.094 | 0.958 ± 0.023 |
| rmsprop-eope | 0.765 ± 0.216 | 0.960 ± 0.017 | 0.854 ± 0.187 | 0.964 ± 0.016 | 0.976 ± 0.011 |
| deep-eope | 0.602 ± 0.279 | 0.701 ± 0.230 | 0.528 ± 0.300 | 0.749 ± 0.209 | 0.785 ± 0.259 |
| devnet | – | 0.557 ± 0.104 | 0.594 ± 0.111 | 0.698 ± 0.163 | 0.812 ± 0.164 |
| feawad | – | 0.862 ± 0.088 | 0.913 ± 0.069 | 0.892 ± 0.101 | 0.937 ± 0.083 |
| deep-sad | 0.803 ± 0.236 | 0.868 ± 0.182 | 0.942 ± 0.022 | 0.943 ± 0.069 | 0.968 ± 0.007 |
| pro | – | 0.726 ± 0.179 | 0.728 ± 0.163 | 0.870 ± 0.128 | 0.905 ± 0.106 |
| nfad (iaf) | **0.981** ± 0.001 | **0.984** ± 0.002 | **0.993** ± 0.002 | **0.997** ± 0.002 | **0.997** ± 0.002 |
| nfad (nsf) | 0.704 ± 0.007 | 0.875 ± 0.121 | 0.901 ± 0.082 | 0.926 ± 0.041 | 0.945 ± 0.022 |

**Table 2  ROC AUC on the HIGGS dataset.** 'nfad*' is our algorithm.

|  | one class | 100 | 1000 | 10000 | 1000000 |
|---|---|---|---|---|---|
| rae-oc | 0.531 ± 0.000 | 0.531 ± 0.000 | 0.531 ± 0.000 | 0.531 ± 0.000 | 0.531 ± 0.000 |
| deep-svdd-oc | 0.513 ± 0.000 | 0.513 ± 0.000 | 0.513 ± 0.000 | 0.513 ± 0.000 | 0.513 ± 0.000 |
| two-class | – | 0.504 ± 0.017 | 0.529 ± 0.007 | 0.566 ± 0.006 | 0.858 ± 0.002 |
| dae | – | 0.502 ± 0.003 | 0.522 ± 0.003 | 0.603 ± 0.002 | 0.745 ± 0.005 |
| brute-force-ope | 0.508 ± 0.000 | 0.500 ± 0.009 | 0.520 ± 0.003 | 0.572 ± 0.005 | 0.859 ± 0.001 |
| hmc-eope | 0.509 ± 0.000 | 0.523 ± 0.005 | 0.567 ± 0.008 | 0.648 ± 0.005 | 0.848 ± 0.001 |
| rmsprop-eope | 0.503 ± 0.000 | 0.506 ± 0.008 | 0.531 ± 0.008 | 0.593 ± 0.011 | **0.861** ± 0.000 |
| deep-eope | 0.531 ± 0.000 | 0.537 ± 0.011 | 0.560 ± 0.008 | 0.628 ± 0.005 | 0.860 ± 0.001 |
| devnet | – | 0.565 ± 0.011 | **0.697** ± 0.006 | **0.748** ± 0.004 | 0.748 ± 0.003 |
| feawad | – | 0.551 ± 0.009 | 0.555 ± 0.014 | 0.554 ± 0.020 | 0.549 ± 0.018 |
| deep-sad | 0.502 ± 0.010 | 0.511 ± 0.006 | 0.561 ± 0.016 | 0.740 ± 0.011 | 0.833 ± 0.002 |
| pro | – | 0.533 ± 0.022 | 0.569 ± 0.011 | 0.570 ± 0.012 | 0.582 ± 0.015 |
| nfad (iaf) | **0.572** ± 0.009 | **0.574** ± 0.008 | 0.586 ± 0.009 | 0.623 ± 0.007 | 0.750 ± 0.008 |
| nfad (nsf) | 0.531 ± 0.010 | 0.519 ± 0.008 | 0.554 ± 0.009 | 0.659 ± 0.007 | 0.807 ± 0.007 |

Also, hyperparameters like Jacobian regularization $\lambda$ and tail size $p$ must be accurately chosen. This fact is illustrated in Figs. 1 and 2, where we show the different samples quality and the performance of our algorithm for different hyperparameters values. To find suitable values, some heuristics can be used. For instance, optimal tail location $p$ can be estimated based on known anomalies from the training dataset, whereas Jacobian regularization $\lambda$ in the NF training process can be linearly scheduled like KL factor in (*Hasan et al., 2020*).

On tabular data (Tables 1, 2 and 3), the proposed NFAD method shows statistically significant improvement over other AD algorithms in many experiments, where the amount of anomalous samples is extremely low.

**Table 3  ROC AUC on the SUSY dataset.** 'nfad*' is our algorithm.

|  | one class | 100 | 1000 | 10000 | 1000000 |
|---|---|---|---|---|---|
| rae-oc | 0.586 ± 0.000 | 0.586 ± 0.000 | 0.586 ± 0.000 | 0.586 ± 0.000 | 0.586 ± 0.000 |
| deep-svdd-oc | 0.568 ± 0.000 | 0.568 ± 0.000 | 0.568 ± 0.000 | 0.568 ± 0.000 | 0.568 ± 0.000 |
| two-class | – | 0.652 ± 0.031 | 0.742 ± 0.011 | 0.792 ± 0.004 | 0.878 ± 0.000 |
| dae | – | 0.715 ± 0.020 | 0.766 ± 0.009 | 0.847 ± 0.002 | 0.876 ± 0.000 |
| brute-force-ope | 0.597 ± 0.000 | 0.672 ± 0.020 | 0.748 ± 0.012 | 0.792 ± 0.003 | 0.878 ± 0.000 |
| hmc-eope | 0.528 ± 0.000 | 0.738 ± 0.019 | 0.770 ± 0.012 | 0.816 ± 0.006 | 0.877 ± 0.000 |
| rmsprop-eope | 0.528 ± 0.000 | 0.714 ± 0.019 | 0.760 ± 0.016 | 0.807 ± 0.004 | 0.877 ± 0.000 |
| deep-eope | 0.652 ± 0.000 | 0.670 ± 0.054 | 0.746 ± 0.024 | 0.813 ± 0.003 | 0.878 ± 0.000 |
| devnet | – | 0.747 ± 0.023 | 0.849 ± 0.002 | 0.853 ± 0.002 | 0.854 ± 0.004 |
| feawad | – | 0.758 ± 0.019 | 0.760 ± 0.028 | 0.760 ± 0.022 | 0.762 ± 0.025 |
| deep-sad | 0.534 ± 0.022 | 0.581 ± 0.027 | 0.785 ± 0.014 | 0.860 ± 0.009 | 0.872 ± 0.008 |
| pro | – | **0.833** ± 0.008 | **0.861** ± 0.002 | 0.863 ± 0.001 | 0.863 ± 0.002 |
| nfad (iaf) | 0.701 ± 0.007 | 0.801 ± 0.007 | 0.829 ± 0.007 | **0.868** ± 0.006 | **0.880** ± 0.000 |
| nfad (nsf) | **0.785** ± 0.001 | 0.811 ± 0.013 | 0.855 ± 0.012 | 0.865 ± 0.001 | 0.876 ± 0.003 |

**Table 4  ROC AUC on the MNIST dataset.** 'nfad*' is our algorithm.

|  | one class | 1 | 2 | 4 |
|---|---|---|---|---|
| nn-oc | 0.787 ± 0.139 | 0.787 ± 0.139 | 0.787 ± 0.139 | 0.787 ± 0.139 |
| rae-oc | **0.978** ± 0.017 | **0.978** ± 0.017 | 0.978 ± 0.017 | 0.978 ± 0.017 |
| deep-svdd-oc | 0.641 ± 0.086 | 0.641 ± 0.086 | 0.641 ± 0.086 | 0.641 ± 0.086 |
| two-class | – | 0.879 ± 0.108 | 0.957 ± 0.050 | 0.987 ± 0.014 |
| dae | – | 0.934 ± 0.035 | 0.964 ± 0.032 | 0.984 ± 0.012 |
| brute-force-ope | 0.783 ± 0.120 | 0.915 ± 0.096 | 0.968 ± 0.041 | 0.986 ± 0.015 |
| hmc-eope | 0.694 ± 0.167 | 0.933 ± 0.060 | 0.974 ± 0.023 | 0.989 ± 0.011 |
| rmsprop-eope | 0.720 ± 0.186 | 0.933 ± 0.062 | 0.977 ± 0.023 | 0.990 ± 0.009 |
| deep-eope | 0.793 ± 0.129 | 0.942 ± 0.048 | **0.979** ± 0.016 | **0.991** ± 0.007 |
| deep-sad | 0.636 ± 0.114 | 0.859 ± 0.094 | 0.908 ± 0.071 | 0.947 ± 0.059 |
| pro | – | 0.911 ± 0.096 | 0.944 ± 0.065 | 0.952 ± 0.079 |
| nfad (resflow) | 0.682 ± 0.115 | 0.909 ± 0.959 | 0.935 ± 0.111 | 0.972 ± 0.019 |

On image data (Tables 4, 5 and 6), the proposed method shows competitive quality along with other state-of-the-art AD methods, significantly outperforming the existing algorithms on CIFAR dataset.

Our experiments suggest the main reason for the proposed method to have lower performance with respect to others on image data is a tendency of normalizing flows to estimate the likelihood of images by its local features instead of common semantics, as described by *Kirichenko, Izmailov & Wilson (2020)*. We also find that the overfitting of the classifier must be carefully monitored and addressed, as this might lead to the deterioration of the algorithm.

However, the results obtained on HIGGS, KDD, SUSY and CIFAR-10 datasets demonstrated the big potential of the proposed method over previous AD algorithms.

**Table 5  ROC AUC on the CIFAR-10 dataset.** 'nfad*' is our algorithm.

|  | one class | 1 | 2 | 4 |
|---|---|---|---|---|
| nn-oc | $0.532 \pm 0.101$ | $0.532 \pm 0.101$ | $0.532 \pm 0.101$ | $0.532 \pm 0.101$ |
| rae-oc | $0.585 \pm 0.126$ | $0.585 \pm 0.126$ | $0.585 \pm 0.126$ | $0.585 \pm 0.126$ |
| deep-svdd-oc | $0.546 \pm 0.058$ | $0.546 \pm 0.058$ | $0.546 \pm 0.058$ | $0.546 \pm 0.058$ |
| two-class | – | $0.659 \pm 0.093$ | $0.708 \pm 0.086$ | $0.748 \pm 0.082$ |
| dae | – | $0.587 \pm 0.109$ | $0.634 \pm 0.109$ | $0.671 \pm 0.093$ |
| brute-force-ope | $0.540 \pm 0.101$ | $0.688 \pm 0.087$ | $0.719 \pm 0.079$ | $0.757 \pm 0.073$ |
| hmc-eope | $0.547 \pm 0.116$ | $0.678 \pm 0.091$ | $0.709 \pm 0.084$ | $0.739 \pm 0.074$ |
| rmsprop-eope | $0.565 \pm 0.111$ | $0.678 \pm 0.081$ | $0.715 \pm 0.083$ | $0.746 \pm 0.069$ |
| deep-eope | $0.564 \pm 0.094$ | $0.674 \pm 0.100$ | $0.690 \pm 0.092$ | $0.719 \pm 0.099$ |
| deep-sad | $0.532 \pm 0.061$ | $0.653 \pm 0.072$ | $0.680 \pm 0.069$ | $0.689 \pm 0.065$ |
| pro | – | $0.635 \pm 0.081$ | $0.653 \pm 0.075$ | $0.670 \pm 0.069$ |
| nfad (resflow) | $\mathbf{0.597} \pm 0.083$ | $\mathbf{0.800} \pm 0.095$ | $\mathbf{0.863} \pm 0.042$ | $\mathbf{0.877} \pm 0.045$ |

**Table 6  ROC AUC on the Omniglot dataset. Note that for this task only Greek, Futurama and Braille alphabets were considered as normal classes.** 'nfad*' is our algorithm.

|  | one class | 1 | 2 | 4 |
|---|---|---|---|---|
| nn-oc | $0.521 \pm 0.166$ | $0.521 \pm 0.166$ | $0.521 \pm 0.166$ | $0.521 \pm 0.166$ |
| rae-oc | $0.771 \pm 0.221$ | $0.771 \pm 0.221$ | $0.771 \pm 0.221$ | $0.771 \pm 0.221$ |
| deep-svdd-oc | $0.640 \pm 0.153$ | $0.640 \pm 0.153$ | $0.640 \pm 0.153$ | $0.640 \pm 0.153$ |
| two-class | – | $0.799 \pm 0.162$ | $0.862 \pm 0.115$ | $0.855 \pm 0.125$ |
| dae | – | $0.737 \pm 0.134$ | $0.821 \pm 0.104$ | $0.805 \pm 0.121$ |
| brute-force-ope | $0.503 \pm 0.213$ | $0.724 \pm 0.222$ | $0.765 \pm 0.208$ | $0.825 \pm 0.126$ |
| hmc-eope | $0.710 \pm 0.178$ | $0.801 \pm 0.139$ | $0.842 \pm 0.112$ | $0.842 \pm 0.115$ |
| rmsprop-eope | $0.678 \pm 0.274$ | $0.821 \pm 0.143$ | $0.855 \pm 0.112$ | $0.863 \pm 0.111$ |
| deep-eope | $0.696 \pm 0.172$ | $0.808 \pm 0.140$ | $0.851 \pm 0.110$ | $0.842 \pm 0.122$ |
| deep-sad | $\mathbf{0.832} \pm 0.123$ | $\mathbf{0.856} \pm 0.123$ | $\mathbf{0.885} \pm 0.095$ | $\mathbf{0.884} \pm 0.091$ |
| pro | – | $0.750 \pm 0.160$ | $0.765 \pm 0.163$ | $0.787 \pm 0.153$ |
| nfad (resflow) | $0.567 \pm 0.108$ | $0.727 \pm 0.188$ | $0.868 \pm 0.111$ | $0.870 \pm 0.102$ |

With the advancement of new ways of NF application to images, the results are expected to improve for this class of datasets as well. In particular, we believe our method to be widely applicable in the industrial environment, where the task of AD can take advantage of both tabular and image-like datasets.

It also should be emphasized that unlike state-of-the-art AD algorithms (*Pang et al., 2019*; *Zhou et al., 2021*; *Ruff et al., 2019*), we propose a model-agnostic data augmentation algorithm that does not modify AD model training scheme and architecture. It enriches the input training anomalies set requiring only normal samples in the augmentation process (Fig. 3).

```
Classifier: DNN(
     (0): Linear(in_features=D, out_features=3*D, bias=True)
     (1): ReLU()
     (2): Linear(in_features=3*D, out_features=2*D, bias=True)
     (3): ReLU()
     (4): Linear(in_features=2*D, out_features=1, bias=True)
  )
```

**Figure 4  Tabular data classifier architecture.**

## CONCLUSION

In this work, we present a new model-agnostic anomaly detection training scheme that deals efficiently with hard-to-address problems both by one-class or two-class methods. The solution combines the best features of one-class and two-class approaches. In contrast to one-class approaches, the proposed method makes the classifier effectively utilize any number of known anomalous examples, but, unlike conventional two-class classification, does not require an extensive number of anomalous samples. The proposed algorithm significantly outperforms the existing anomaly detection algorithms in most realistic anomaly detection cases. This approach is especially beneficial for anomaly detection problems, in which anomalous data is non-representative, or might drift over time.

The proposed method is fast, stable and flexible both in terms of training and inference stages; unlike previous methods, any classifier can be used in the scheme with any number of anomalies in the training dataset. Such a universal augmentation scheme opens wide prospects for further anomaly detection study and makes it possible to use any classifier on any kind of data. Also, the results on datasets with images are improvable with new techniques of normalizing flows become available.

## APPENDIX A. TRAIN AND IMPLEMENTATION DETAILS

All the code is implemented using the PyTorch (*Paszke et al., 2019*) framework. For augmentation, Resflow (*Chen et al., 2019*), NSF (*Durkan et al., 2019*) and IAF (*Kingma et al., 2016*) are trained with default parameters. As a classifier, a dense classifier with three layers is used for tabular data (see Fig. 4) and built-in ResFlow classification head is used for images. Tabular data classifier is trained 10 epochs with batch size 100 using AdamW (*Loshchilov & Hutter, 2017*) optimizer with default PyTorch parameters. For image data, ResFlow classification head is trained 8 epochs with batch size 40 using Adam (*Kingma & Ba, 2014*) optimizer with default PyTorch parameters.

### Funding

The research leading to these results has received funding from Russian Science Foundation under grant agreement no. 19-71-30020. The research was also supported through

computational resources of HPC facilities at NRU HSE. The funders had no role in study design, data collection and analysis, decision to publish, or preparation of the manuscript.

### Grant Disclosures

The following grant information was disclosed by the authors:
Russian Science Foundation: No. 19-71-30020.
NRU HSE.

### Competing Interests

The authors declare there are no competing interests.

### Author Contributions

- Artem Ryzhikov conceived and designed the experiments, performed the experiments, analyzed the data, performed the computation work, prepared figures and/or tables, authored or reviewed drafts of the paper, and approved the final draft.
- Maxim Borisyak conceived and designed the experiments, performed the experiments, analyzed the data, performed the computation work, authored or reviewed drafts of the paper, and approved the final draft.
- Andrey Ustyuzhanin and Denis Derkach conceived and designed the experiments, authored or reviewed drafts of the paper, and approved the final draft.

### Data Availability

The code is available at GitLab: https://gitlab.com/lambda-hse/nfad.

The data is available at:

- Moons dataset: https://scikit-learn.org/stable/modules/generated/sklearn.datasets.make_moons.html
- SUSY dataset: https://archive.ics.uci.edu/ml/datasets/SUSY
- HIGGS dataset: https://archive.ics.uci.edu/ml/datasets/HIGGS
- MNIST dataset: http://yann.lecun.com/exdb/mnist/
- CIFAR dataset: https://www.cs.toronto.edu/~kriz/cifar.html
- KDD dataset: http://kdd.ics.uci.edu/databases/kddcup99/kddcup99.html
- OMNIGLOT dataset: https://github.com/brendenlake/omniglot

### Supplemental Information

Supplemental information for this article can be found online at http://dx.doi.org/10.7717/peerj-cs.757#supplemental-information.

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
