# Peer review of "NFAD: fixing anomaly detection using normalizing flows"

_PeerJ Computer Science, doi:10.7717/peerj-cs.757_

## Round 0.1 · original submission · Major Revisions

· Academic Editor

Major Revisions

Please revise the manuscript following the comments from the reviewers to be considered.

·

Basic reporting

The introduction could be more detailed, such as adding the reason why the proposed method in the manuscript works.

I suggest that the authors add more recent and closely related references on the same problem, such as:

(1) Feature Encoding with AutoEncoders for Weakly-supervised Anomaly Detection, IEEE Transactions on Neural Networks and Learning Systems, 2021.
(2) Deep Weakly-supervised Anomaly Detection. arXiv preprint: 1910.13601 (2019).

I also encourage the authors to consider comparing their performance with those of the two citations I recommended. The authors had use an early version of the second recommended citation as a comparing method(devnet); the first citation has a superior performance than that of devnet on the same problem.

The first paper is my own paper. And the second one has no conflict of interest with me.

The manuscript is not well formated in the current version. Also, there are some typos and errors in the content, e.g., the caption of some tables are wrongly represented as "Figure X.".

Excludingly, this manuscript basically meets the requirements.

Experimental design

In the experiments, the analysis of influencing factors is missing, and the author can add some experiments to this regard.

Validity of the findings

More experiments are suggested to conduct to show the effect of the proposed method as well as the major factors that may have impact on its performance.

Additional comments

I attempted to run the code, but an error occured. Maybe there's something wrong with my environment or my network. But I hope the authors make sure the code could run automatically without any additional change.

Also for this code, the authors are suggested to provide more detailed instructions.

Reviewer 2 ·

Basic reporting

Clear and unambiguous, professional English used throughout.
The language is correct and the article is written in clear and professional English.
Literature references, sufficient field background/context provided.
The literature on related papers could be more explored. Just a few numbers of papers were cited, and the article mentions just a few methods of AD, specially based on classification, while there are others as ones based on Clustering, distance and density (as in F. Meng, G. Yuan, S. Lv, Z. Wang, and S. Xia, “An overview on trajectory outlier detection,” Artificial Intelligence Review, vol. 52, no. 4, pp. 2437–2456, 2019).
Professional article structure, figures, tables. Raw data shared.
The structure doesn’t follow the format suggest by the journal. It combines “Results” and “Discussion” in just one item, what makes the paper less clear, because there are discussions presented before results. Result are present in 3 paragraphs after Figure 4 and before figures 7 to 9.
In Figure 1, specially the last diagram is too hard to see the tail p-values. Figures 4, 5, 6, 7, 8 and 9 are not figures but table. They should shave been mentioned as tables not figures.
Figures 8 and 9 are misplaced in the text. They are in “Conclusion”, but they should be included in “Results and Discussions” as figures 4 to 7 are.
In figure 7 and 8, the best results are bolded as in other figures. They should have been bolded despite the best results are not provided by the proposed algorithm.
All appropriate raw data have been made available in accordance with our Data Sharing policy.
Self-contained with relevant results to hypotheses.
The article proposes a new technique to address DA problem. The problems are well stated, and the hypothesis is tested, showing relevant results, even the ones that the proposed algorithm doesn’t performance well.
The submission is ‘self-contained,’ and represents an appropriate ‘unit of publication’. It also includes all results relevant to the hypothesis.
Formal results should include clear definitions of all terms and theorems, and detailed proofs.
In topic 3, when authors state that normalizing flows generative mode aims to fit the exact probability distribution of data, they don’t mention that normalizing flows provides exact inference and log-likelihood evaluation as its merits (D. P. Kingma and P. Dhariwal, “Glow: Generative flow with invertible 1x1 convolutions,” in Advances in Neural Information Processing Systems, 2018, pp. 10 215–10 224). It is important because when the model is trained in topic 4, it is trained by standard for normalizing scheme of maximization the log-likelihood, which is not described previously when normalizing flows are described (Topic 3).

Experimental design

Original primary research within Aims and Scope of the journal.
The submission is within Aims and Scope of the journal, because proposes a new algorithm to AD task, a field related to Computer Sciences. The paper is a Research Article, in which a well stated hypothesis is tested and the results of these tests are presented.
Research question well defined, relevant & meaningful. It is stated how research fills an identified knowledge gap.
The research question is well defined and also the article scope. However, in the second line of “Problem Statement” item, the statement “In practice, while such samples are, indeed, most likely to be anomalous, often some
anomalies might not be distinguishable from normal samples” doesn’t have any reference.
Also, the effect of the number of anomaly samples in the training data should be addressed more deeply, as in Exploring normalizing flows for Anomaly Detection (Pathak, C. 2019), as this effect is one of the most important issues in the paper.
Rigorous investigation performed to a high technical & ethical standard.
The investigation must have been conducted rigorously and to a high technical standard. The research must have been conducted in conformity with the prevailing ethical standards in the field.
Methods described with sufficient detail & information to replicate.
Methods should be described with sufficient information to be reproducible by another investigator.

Validity of the findings

Impact and novelty not assessed. Meaningful replication encouraged where rationale & benefit to literature is clearly stated.
Conclusions are based in a metric presented in tables, so we can see clearly where the proposed algorithm surpass others and where it doesn’t. However, the metric employed is not clearly described. We infer that authors use accuracy, but this option is not justified.
All underlying data have been provided; they are robust, statistically sound, & controlled.
The data is provided and it was possible to rerun the experiment.
Conclusions are well stated, linked to original research question & limited to supporting results.
The conclusions are appropriately stated, and connected to the original question investigated, and are supported by the results, as based in obtained results described in figures 4 to 9. The proposed algorithm outperformances many tested algorithms in most of situations, except when dialing with images dataset.

Additional comments

As general conclusion, the paper can be published after mandatory corrections, specially these described in item 1.

Annotated reviews are not available for download in order to protect the identity of reviewers who chose to remain anonymous.

---

## Round 0.2 · Minor Revisions

· Academic Editor

Minor Revisions

Please revise the manuscript following the comment from the reviewer.

·

Basic reporting

It seems that the authors have addressed most my concerns. I suggest to a minor revision for the manuscript in its current form. The comments are as follows:1
1. for the two references, "(1) Feature Encoding with AutoEncoders for Weakly-supervised
Anomaly Detection, IEEE Transactions on Neural Networks and
Learning Systems, 2021. (2) Deep Weakly-supervised Anomaly
Detection. arXiv preprint: 1910.13601 (2019)."
What is the major technical differences between the references and the proposed method? I guess that the authors did not display the bad cases for the proposed method, which the authors should present and highlight them.
2. some references are not complete, e.g., ““Zhou, Y., Song, X., Zhang, Y., Liu, F., Zhu, C., and Liu, L. (2021). Feature encoding with autoencoders for weakly-supervised anomaly detection.””, which lacks the publication title and etc.
"241 Chen, R. T. Q., Behrmann, J., Duvenaud, D., and Jacobsen, J.-H. (2019). Residual flows for invertible
242 generative modeling."
3 a detailed lists of revisions are needed in the response letter for each coments in the previous review, so that the reviewer can see the revisions more clear.

Experimental design

please see the above

Validity of the findings

please see the above

Additional comments

no

---

## Round 0.3 · accepted · Accept

· Academic Editor

Accept

I believe the current version is sufficient to be accepted.